# Açai (*Euterpe oleracea* Mart.) Seed Extract Induces ROS Production and Cell Death in MCF-7 Breast Cancer Cell Line

**DOI:** 10.3390/molecules26123546

**Published:** 2021-06-10

**Authors:** Marcos Antonio Custódio Neto da Silva, Jonas Henrique Costa, Taícia Pacheco-Fill, Ana Lúcia Tasca Gois Ruiz, Flávia Castello Branco Vidal, Kátia Regina Assunção Borges, Sulayne Janaina Araújo Guimarães, Ana Paula Silva de Azevedo-Santos, Kaio Eduardo Buglio, Mary Ann Foglio, Maria do Carmo Lacerda Barbosa, Maria do Desterro Soares Brandão Nascimento, João Ernesto de Carvalho

**Affiliations:** 1Post-Graduate Program in Internal Medicine, Faculty of Medical Science, Universidade Estadual de Campinas, Rua Tessália Vieira de Camargo, 126, Cidade Universitária Zeferino Vaz. CEP, Campinas 13083-887, SP, Brazil; marcos_antonio456@hotmail.com; 2Institute of Chemistry, Universidade Estadual de Campinas, CP 6154, Campinas 13083-970, SP, Brazil; j161206@dac.unicamp.br (J.H.C.); taicia@iqm.unicamp.br (T.P.-F.); 3Faculty of Pharmaceutical Sciences, Universidade Estadual de Campinas, Campinas 13083-859, SP, Brazil; ana.ruiz@fcf.unicamp.br (A.L.T.G.R.); kaiobuglio@gmail.com (K.E.B.); maryann.foglio@fcf.unicamp.br (M.A.F.); 4Post-Graduate Program in Adult Heath, Department of Patology, Federal University of Maranhão (UFMA), São Luís 65080-805, MA, Brazil; flavidaly@yahoo.com.br (F.C.B.V.); kareborges@gmail.com (K.R.A.B.); 5Post-Graduate Program in Health Sicencies, Federal University of Maranhão (UFMA), São Luís 65080-805, MA, Brazil; sulaynebio@hotmail.com (S.J.A.G.); apsazevedo@yahoo.com.br (A.P.S.d.A.-S.); 6Post-Graduate Program in Family Health, Department of Medicine I, Federal University of Maranhão (UFMA), São Luís 65080-805, MA, Brazil; carminha13032009@hotmail.com

**Keywords:** *Euterpe oleracea* Mart., flavonoids, breast cancer, autophagy, reactive oxygen species, mass spectrometry

## Abstract

*Euterpe oleracea* Mart. (açai) is a native palm from the Amazon region. There are various chemical constituents of açai with bioactive properties. This study aimed to evaluate the chemical composition and cytotoxic effects of açai seed extract on breast cancer cell line (MCF-7). Global Natural Products Social Molecular Networking (GNPS) was applied to identify chemical compounds present in açai seed extract. LC-MS/MS and molecular networking were employed to detect the phenolic compounds of açai. The antioxidant activity of açai seed extract was measured by DPPH assay. MCF-7 breast cancer cell line viability was evaluated by MTT assay. Cell death was evaluated by flow cytometry and time-lapse microscopy. Autophagy was evaluated by orange acridin immunofluorescence assay. Reactive oxygen species (ROS) production was evaluated by DAF assay. From the molecular networking, fifteen compounds were identified, mainly phenolic compounds. The açai seed extract showed cytotoxic effects against MCF-7, induced morphologic changes in the cell line by autophagy and increased the ROS production pathway. The present study suggests that açai seed extract has a high cytotoxic capacity and may induce autophagy by increasing ROS production in breast cancer. Apart from its antioxidant activity, flavonoids with high radical scavenging activity present in açai also generated NO (nitric oxide), contributing to its cytotoxic effect and autophagy induction.

## 1. Introduction

*Euterpe oleracea* Mart., a native palm from the Amazon region, popularly known as açai, açai do Pará and juçara, is a multi-stem palm, with up to 25 stems per clump. The trunks in adults have heights ranging between 3 m and 20 m and a diameter of 7 cm to 18 cm [1]. Each stem holds, at its end, a set of 8–14 compounds, pinnate leaves and spiral arrangement, with 40–80 pairs of leaflets, opposite or sub-opposite. The fruit ripening is complete in about 175 days, presenting a violet color and diameter of about 13.5 mm in Henderson (1995) [2].

Açai is considered a functional food. Due to its many biological activities, the consumption of açai is increasing outside of the Amazon [3]. Açai has also been applied in the cosmetic and pharmaceutical industries [4,5,6,7]. Among the previously studied activities, the antioxidant and anti-inflammatory effects are the most commonly reported. Assessments of the significant antioxidant capacity of açai pulp in free radicals have been published. Results show high antioxidant activity against the DPPH radical, anion superoxide, peroxyl radicals, hydroxyl radicals and the inhibition of oxidation of liposomes [8,9]. As can be seen, consumption of açai juice or pulp by healthy human volunteers has shown antioxidant capacity in vivo [10].

There are various chemical constituents of açai with bioactive properties. Previous phytochemical analysis revealed the presence of flavonoids, anthocyanins, benzenoid lignans, benzoquinone, monoterpenoids, norisoprenoids, as well as essential fatty acids [9,11,12]. The fruit consists of protein (9.1%), lipids (42.3%), carbohydrates (43.4%), vitamins B1, C and E and minerals such as iron, phosphorus, calcium, potassium, starch, fiber [13]. The major anthocyanins in açai are cyanidin 3-glucoside and cyanidin 3-rutinoside [8,12,14,15,16,17]. The major flavonoids found in açai are quercetin, orientin and its derivatives as well as proanthocyanidins. [7,9,14].

Açai showed antitumoral functions due to its anti-inflammatory, antiproliferative and pro-apoptotic properties [18]. In vitro studies showed that açai decreased cell viability and induced apoptosis in glioma cells [19], breast cancer cells [20,21] and colon cancer cells [22]. Açai seed extract induced cell cycle arrest and apoptosis in lung cancer cells [23]. Experimental models of cancer in vivo also evidenced anticarcinogenic and chemopreventive effects of açai [24,25,26,27,28,29]

GNPS is a modern platform for analyzing and storing MS/MS spectra [30]. GNPS can be used for molecular networking, which is a spectral correlation and visualization approach that can detect sets of spectra from related molecules [31,32,33], even when the spectra themselves are not matched to any known compounds [34].

Preliminary reports of our research group have evidenced the cytotoxic effects of açai seed extract in breast cancer cell line [20,21].

Considering the diversity of chemical compounds present in açai, this study aimed to identify the compounds present in the pulp, seed and total fruit of *Euterpe oleracea* Mart. and to better evaluate the cytotoxic pathway of açai seed extract in MCF-7 breast cancer cell line.

## 2. Materials and Methods

### 2.1. Preparation of Lyophilized Hydro-Alcoholic Extract of the Total Fruit, Pulp and Seed of Euterpe oleracea Mart

The fruits of açai (*Euterpe oleracea* Mart.) used in this study came from the Juçara Park (São Luís, Maranhão, Brazil). A sample of the specimen was stored under exsiccation at State University of Maranhão (UEMA) and deposited with the World International Property Organization under registration number PI0418614-1.

The fruits were previously conditioned under refrigeration at −20 °C in the Laboratory of Cell Culture of the Nucleus of Basic and Applied Immunology of the Federal University of Maranhão (UFMA). After thawing at room temperature, the sample was separated into three parts: seed, pulp and total fruit (seed + pulp). The extraction process followed the methodology developed by Moura et al. (2012) [35].

Approximately 360 g of each different portion of açai (seed, pulp and total fruit) was washed in tap water and boiled in distilled water for 5 to 10 min. Subsequently, the portions were ground and then homogenized with 400 mL of ethanol under stirring for 2 h. The resulting extracts were stored at 4 °C and protected from light for 10 days. After this maturation period, the hydro-alcoholic extracts were filtered and the liquid phase concentrated in a rotary evaporator at approximately 40 °C and then lyophilized at a temperature of −30 to −40 °C and a vacuum of 200 mm Hg.

The extracts were kept at −20 °C until the day of use.

### 2.2. Thin Layer Chromatography (TLC) Analysis

The hydro-alcoholic açai seed extract was preliminarily analyzed by TLC using silica gel 60F_254_ in an aluminum chromatography sheet (20 cm × 20 cm × 0.15 mm; Merck^®^, Darmstadt, Germany). As a mobile phase, the mixture of solvents: n-butanol: glacial acetic acid: water (40:10:50 *v/v/v*—upper phase) was used. After elution, chromatogram was sprayed with natural products reagent (NP/PEG) and observed under ultraviolet light at wavelength 254 and 365 nm [36].

### 2.3. MS/MS Analysis

Crude extracts of seed, pulp and total fruit of açai (*Euterpe oleracea* Mart.) were suspended in 2 mL of MeOH for HPLC and centrifuged at 13,000 rpm for 5 min. Next, 100 µL of the supernatant solution was filtered at 0.22 µm and diluted into 900 µL MeOH HPLC.

The samples were analyzed using an LC Agilent 1200 mass spectrometer (Agilente, Santa Clara, CA, USA) coupled with Agilent iFunnel 6550 Q-ToF LC/MS. The electrospray ionization was operated in positive mode. Fragmentor voltage and collision energies were selected following operating conditions according to each analysis during infusions. Five precursors per cycle were selected. Stationary phase: Thermo Scientific column Accucore C18 2.6 µm, 2.1 mm × 100 mm. Mobile phase: acetonitrile and 0.1% formic acid. Flow rate: 0.2 mL min^−1^. Organic phase in gradient mode from 5% to 98% within 10 min, hold for 5 min, up to 5% within 1.2 min and hold for 4.8 min. Total run time: 21 min. Injection volume: 2 µL.

Agilent Mass Hunter Workstation Software B06.00 was used to process the spectra.

### 2.4. MS/MS Molecular Networking

A molecular network for *E. oleracea* Mart. was created using the online workflow at GNPS (http://gnps.ucsd.edu, accessed on 9 June 2021). The data were filtered by removing all MS/MS peaks within ±17 Da of the precursor *m/z*. Tandem mass spectra were window filtered by choosing only the top 6 peaks in the ±50 Da window throughout the spectrum. The data were then clustered with MS-Cluster with a parent mass tolerance of 2.0 Da and an MS/MS fragment ion tolerance of 0.5 Da to create consensus spectra. Further, consensus spectra that contained less than 2 spectra were discarded. A network was then created where edges were filtered to have a cosine score above 0.7 and more than 6 matched peaks. Further edges between two nodes were kept in the network only if each of the nodes appeared in each other’s respective top 10 most similar nodes. The spectra in the network were then searched against GNPS’ spectral libraries. The library spectra were filtered in the same manner as the input data. All matches kept between network spectra and library spectra were required to have a score above 0.7 and at least 6 matched peaks. MS/MS spectra were networked and visualized with Cytoscape v.3.7.1. (The Cytoscape Consortium, New York, NY, USA).

### 2.5. Antioxidant Activity Assay by DPPH (2,2-Diphenyl-1-picrylhydrazly)

For DPPH assay, aliquots of açai seed extract were mixed with 2.5 mL DPPH methanolic solution (0.06 mM) and allowed to react for half an hour, in the dark. Measurements were performed at 515 nm applying a Turner^®^ 340 spectrophotometer (Barnstead/Thermolyne, Dubuque, IA, USA). The analysis was performed in triplicate and the decline in the DPPH radical absorbance concentration caused by the samples was measured. The results are expressed as % of reduction of radical absorbance according to this equation.
Antioxidant activity (%) = [1 − (sample absorbance/control absorbance) × 10.(1)

A blank sample was prepared using ethanol instead of extract. Trolox standard curve was provided as positive control.

### 2.6. Cell Viability Assay

MCF-7 (ATCC^®^, HTB-22™), derived from human mammary adenocarcinoma and human cells derived from human fibroblast (GM cells), were obtained from the Rio de Janeiro CellBank, which certified their identity and quality (INMETRO, Rio de Janeiro, RJ, Brazil). Cells were grown in Dulbecco’s Modified Eagle’s Medium (DMEM) (Invitrogen) supplemented with fetal bovine serum 10%, penicillin G (60 mg/L), streptomycin (100 mg/L) at 37 °C in 5% of CO_2_.

The samples were diluted in stock solutions in dimethyl sulfoxide (DMSO) (Merck^®^) at a concentration of 0.1 g/mL. MCF-7 (1 × 10^4^ cell/mL) cultured cells were treated with 0.25, 2.5, 25 and 250 μg/mL of açai seed extract for 24, 48 and 72 h. GM (1 × 10^4^ cell/mL) cultured cells were treated with 7.8 to 1000 μg/mL of açai seed extract for 24 h. Six wells were included for control (DMEM).

The supernatant was removed, and 100 μL of 3-(4,5-dimethylthiazol-2-yl)-2,5-diphenyltetrazolium bromide (MTT) was added to each well and incubated at 37 °C. After this, cells were washed 3 times with PBS and were incubated for 3 h with protection from light and, after washing, the formazan crystals were solubilized with DMSO (100 μL) and the absorbance measured at 540 nm was measured using a Spectra Max 190 spectrophotometer (Molecular Devices, Sunnyvale, CA, USA).

### 2.7. Clonogenicity Assay

MCF-7 cells (7 × 10^2^ cells/mL) were treated with 10 and15 μg/mL of açai seed extract and cultured for 15 days in 12-well plates to determine the effects of the treatment on the clonogenic potential of the cells. The formed colonies were fixed with ethanol (200 μL) for 10 min, stained with a crystal violet solution (0.05% crystal violet and 20% ethanol) for 10 min, washed twice with distilled water, and solubilized with acetic acid 33%, after which the 595 nm absorbance was measured using a Spectra Max 190 spectrophotometer.

### 2.8. Quantification of Apoptosis by Flow Cytometry

In order to evaluate death mechanisms induced by açai seed extract, the apoptotic cells were stained using Vybrant^®^ FAM Caspase-3 and -7 Assay Kit (Molecular Probes, Eugene, OR, USA) according to manufacturer’s instructions. Cells were treated with 25 and 250 μg/mL of açai seed extract for 6 and 24 h. After this, cells were harvested and resuspended to a concentration of 1 × 10^6^ cells/mL in fresh culture medium. Aliquots (300 μL) of cell suspensions were transferred to flow tubes and 10 μL of a fluorescent inhibitor of caspase (FLICA) labeling solution was added to each tube except for the negative control. The cells were incubated for 1 h at 37 °C, washed twice, and resuspended in 400 μL wash buffer. Propidium iodide (PI) was added, and the tubes were incubated for 5–10 min on ice before analysis with FACSCalibur flow cytometer (Beckton Dickinson, San Jose, CA, USA). Fluorescence-1 and fluorescence-3 channels were used to detect the signals arriving from FLICA and PI, respectively.

Cells labeled with Vybrant FAM Caspase-3 and -7 Assay Kit were analyzed by flow cytometry using 488 nm excitation and emission filters appropriate for Alexa Fluor^®^ dye (FAM signal) and Texas Red^®^ dye (propidium iodide signal)

### 2.9. DAF-2DA Diacetate Analysis for Reactive Oxygen Species (ROS) Production by Flow Cytometry

To measure ROS levels, cells were treated with 25 and 250 μg/mL of açai seed extract for 6 and 24 h and then washed twice with PBS. The cells were then stained with 10 μM of 4,5-diaminofluorescein-2 diacetate (DAF-2DA, Molecular Probes, Eugene, OR, USA) for 20 min at room temperature in the dark.

The green fluorescence emitted by DAF-2DA was recorded at 515 nm using a flow cytometer (Becton Dickinson, San Jose, CA, USA), and 10,000 events were counted per sample [37].

### 2.10. Time-Lapse Microscopy

Samples were examined in the National Institute of Science and Technology on Photonics Applied to Cell Biology (INFABIC) at the State University of Campinas, using APOTOME Inverted Fluorescence Microscope for Time-Lapse (Carl Zeiss AG, Germany) equipped with a 5×, 0.15 N.A. phase-contrast objective, a custom-stage incubator capable of housing up to four 35 mm Petri dishes and in vitro software 3.2 (Media Cybernetics Inc., Bethesda, MD, USA). Each experiment was repeated a total of three times, resulting in a total of 48 image sequences (12 phase-contrast time-lapse microscopy image sequences per treatment group). Images were acquired at a frequency of every 5 minutes over a course of 24 h. Microscope images were 1392 × 1040 pixels with a resolution of 1.3 μm/pixel.

### 2.11. Acridine Orange Staining

Acridine orange (AO) is a lysotropic dye that accumulates in acidic organelles in a pH-dependent manner and is commonly used to identify acidic vesicular organelles (AVOs) [38]. Under AO staining, the cytoplasm and nucleoli fluoresce was green, whereas in the acidic compartments, such as lysosomes or autophagolysosomes, fluoresce was bright-red or orange-red with blue-light excitation [39].

The MCF-7 cells with or without the 25 μg/mL of açai seed extract treatment were seeded in 3 cm glass bottom dishes for 24 h and were exposed to AO staining medium (1 μg/mL in complete medium) for 45 min. The AO-staining medium was refreshed prior to the subsequent experiments. All the treatment processes were performed with avoidance of environmental light. The dishes were immediately wrapped with aluminum foil and subjected to fluorescent imaging. Samples were examined in the National Institute of Science and Technology on Photonics Applied to Cell Biology (INFABIC) at the State University of Campinas, using APOTOME Inverted Fluorescence Microscope (Carl Zeiss AG, Germany) equipped with a 130 W fluorescence light source. The filter block was used containing a 465 to 495 nm bandpass excitation filter, 505 nm dichroic mirror and 590 nm long-pass barrier filter.

### 2.12. Statistical Analysis

GraphPad Prism 8.4 version for Windows (GraphPad Software, San Diego, CA, USA) was used to perform statistical analysis. The data were analyzed by one-way analysis of variance (ANOVA), followed by T-Student or Dunnett’s post hoc tests. *p* values < 0.05 were considered statistically significant.

## 3. Results

### 3.1. Açai Seed Extract Is Rich in Flavonoids

#### 3.1.1. Thin-Layer Chromatography (TLC) Analysis

Thin-layer chromatography evidenced a great amount of compounds that were further analyzed by LC MS/MMS (Figure 1).

#### 3.1.2. MS/MS Analysis and Molecular Networking

Mass spectrometry data obtained from the analyses of the crude extracts of *E. oleracea* Mart. were used to generate molecular networking clusters at the online platform GNPS (Figure 2).

MS/MS spectra were networked, and such organization enables the visualization of the relationships between identical and related molecules based on the spectral similarity of their MS/MS signatures. An MS/MS cluster, where many nodes are connected by edges, indicates that many related molecules are observed. Furthermore, MS/MS networking enables the visualization of groups possessing unique spectral signatures that indicate that the molecules are distinct from the other molecules in a given dataset. For *E. oleracea* Mart., including seed, pulp and total fruit extracts, based on the MS/MS spectra, similarity was generated with GNPS, which led to the presence of 16 clusters (node ≥ 2) and 132 single nodes as represented in Figure 3.

The results revealed the presence of three distinct phenolic clusters and six phenolic single nodes. GNPS efficiently separated flavonoid glycosides (Cluster I), procyanidins (Cluster II), anthocyanins (Cluster III) and aglycones (Clusters IV and V). Additionally, one single node was identified to be flavonoids: dyhydrokaempferol (VI), nobiletin (VII) and diosmetin (VIII). Epicatechin was also identified (IX) through the GNPS library.

After generating the molecular networks, the node connectivity was visualized. Each node represents one MS/MS spectrum and is labeled with the parent (precursor) mass (Figure 4). Since fragmentation spectra generally reflect the chemical structures of the fragmented ions, it becomes possible to represent the compounds in clusters of similar structures through GNPS platform.

### 3.2. Chemical Compounds Identified by GNPS

GNPS spectral libraries enable dereplication, variable dereplication (approximate matches to spectra of related molecules) and identification of spectra in molecular networks.

For *Euterpe oleracea* Mart. extracts, the molecular network identified 15 compounds, especially flavonoids and anthocyanins. The major compounds were: (−)-epicatechin (**1**), cyanidin 3-glucoside (**2**), procyanidin B2 (**3**), kaempferol-3-*O*-rutinoside (**4**), nobiletin (**5**), dihydrokaempferol (**6**), diosmetin (**7**), 3-*O*-Methylquercetin or isorhamnetin (**8**), isoorientin (**9**), genistein-8-C-glucoside (**10**) and apigenin 6,8-digalactoside (**11**)**.**

The representative chemical structures of each compound are also shown in Table 1.

Other compounds identified were: L-Tryptophan, dioctyl phthalate, melibiose and Pheophorbide A.

The most compounds identified by GNPS spectral libraries were detected in all extracts (seed, pulp and total fruit). Compounds **2**, **4**, **6**, **9** and **11** were identified only in pulp and total fruit extract.

Mass spectral data are described in Appendix A.

### 3.3. Açai Seed Extract Has Antioxidant Activity

Antioxidant activity (%) increased proportionally with extract concentration, reaching 86.8% of the maximum antioxidant activity at a concentration of 500 μg/mL. The EC_50_ value (concentration required to achieve 50% antioxidant activity) was 61.8 μg/mL (Figure 5).

### 3.4. Açai Seed Extract Reduces Cell Viability of MCF-7 Breast Cancer Cell Line

In MCF-7 cell line, açai seed extract strongly reduced the viability of the cells in a time-dependent manner. At 25 and 250 μg/mL concentrations, the extract caused a cytotoxic effect starting at 24 h (*p* < 0.05) which increased after 72 h of treatment when compared to control (Figure 6).

In human cells derived from human fibroblast (GM cells), the açai seed extract was evaluated in different concentrations (7 to 1000 μg/mL). The extract showed cell viability reduction only from 500 μg/mL (Figure 7).

### 3.5. Açai Seed Extract Reduces the Clonogenic Capacity in MCF-7 Cells

In order to evaluate the MCF-7 cell viability reduction after the treatment with açai seed extract, we performed clonogenicity assay. The results of clonogenicity assay showed that the treatment with açai drastically reduced the formation of new colonies (Figure 8A,B).

### 3.6. Açai Seed Extract Induces Morphologic Changes in MCF-7 Breast Cancer Cell Line

The effects of açai seed extract on the morphological features of MCF-7 cells were investigated by time-lapse microscopy. Time-lapse microscopy showed a decrease in cell density in the açai-treated MCF-7 cells (25 μg/mL), as well as cell rounding and shrinking, membrane blebbing and cell lysis with seeming loss of cytoplasmic content starting after 6 h of treatment (Figure 9).

### 3.7. Açai Seed Extract Induces Cell Death by Autophagy in MCF-7 Breast Cancer Cell Line

In order to evaluate cell death in MCF-7 breast cancer cell line, we performed caspase 3 and 7 assay. Treatment with 25 and 250 μg/mL did not show an increase in the percentage of apoptotic cells after 6 and 24 h of treatment (Figure 10A,B).

To confirm this death mechanism, we performed acridine orange assay by immunofluorescence. Treatment with 25 μg/mL of açai seed extract caused an increase in the acidic compartments, such as lysosomes or autophagolysosomes, evidenced by bright-red or orange-red with blue-light excitation (Figure 11).

### 3.8. Açai Seed Extract Increases ROS Production in MCF-7 Breast Cancer Cell Line

Considering that autophagy is the death mechanism of MCF-7 açai-treated cell lines, we used cell-permeable specific fluorescent NO indicator DAF-2DA.

Treatment with 25 μg/mL and 250 μg/mL of açai seed extract caused an increase in ROS production after 6 h when compared to control cells, suggesting the induction of autophagy by ROS increases in açai-treated cells (Figure 12).

## 4. Discussion

GNPS provides a community-led knowledge space in which natural products data can be shared, analyzed and annotated by researchers worldwide. It enables a cycle of annotation in which users curate data, continuous dereplication enables product identification and a knowledge base of reference spectral libraries and public data sets is created [30].

The chemical compounds identified by GNPS hits were mostly flavonoids and anthocyanins. Cyanidin 3-glucoside and cyanidin 3-rutinoside were reported as major anthocyanins [8,12].

The major flavonoids found in açai were quercetin, orientin and its derivatives as well as proanthocyanidins [8,14]. Orientin, homoorientin, vitexin, luteolin, chrysoeriol, quercetin and dihydrokaempferol from açai have also been identified in açai berries [12,40].

Procyanidins and proanthocyanidins were described by Pacheco-Palencia et al. (2008) [41], Del-Pozo Insfran et al. (2004) [17] and Pacheco-Palencia and Talcott (2010) [7]. Quercetin was also identified by Lichtenthaler et al. (2005) [8]. Catechin and epicatechin were identified by Pacheco-Palencia et al. (2009) [21].

This is the first report of the presence of nobiletin and genistein-8-*C*-glucoside in açai (*Euterpe oleracea* Mart.). MS and MS/MS spectral data of all the compounds identified by GNPS and previously identified as components of açai were in agreement with those reported in literature [14]. The percentage of these compounds needs to be further evaluated by HPLC.

Genistein belongs to isoflavones, which is a subclass of flavonoids, a large group of polyphenolic compounds widely distributed in plants. Numerous in vitro studies suggest that isoflavones, particularly genistein, have both chemopreventive and chemotherapeutic potential in multiple tumor types, such as leukemia, lymphoma, prostate, breast, lung and head or neck [42,43]. Like most isoflavones, genistein usually exists in nature in its 7-glycoside or 8-glycoside form, rather than in aglycone form [44].

Nobiletin is also a flavonoid isolated from citrus peels. Recent evidence showed that nobiletin is a multifunctional pharmaceutical agent. The various pharmacological activities include neuroprotection, cardiovascular protection, antimetabolic disorder, anticancer, anti-inflammation and anti-oxidation [45].

The demand for açai is increasing nationally and internationally, with different forms of consumption, because of the great nutritional value and medical properties [46].

Kang et al. (2011) reported that açai is categorized as chemopreventive because of its capacity to reduce the formation of oxygen and nitrogen reactive species [47]. Recent studies have shown that açai can be an antitumor agent by protecting the injured tissue and the carcinogenic activity, but until now, the complete mechanism of action of açai has still been unknown [3,25,26].

Açai seed extract reduced cell viability in MCF-7 breast cancer cell lines. Clonogenic assay also showed that açai seed extract drastically reduced the formation of new colonies.

In order to evaluate cell death, we used morphologic assay and caspase 3/7 assay. Treatment with açai seed extract did not cause apoptosis and there was no difference between the caspase 3 and caspase 7 activity of control and treated cells. However, the acridine orange assay was performed and treatment with 25 μg/mL of açai seed extract caused an increase in the acidic compartments, such as lysosomes or autophagolysosomes.

Silva et al. (2014) evaluated the cytotoxic potential of seed, bark and pulp extract of açai and reported that seed extract was the most effective against MCF-7 cancer cell line [21]. The authors’ reported that açai seed extract induced cell death by autophagy using 20 and 40 μg/mL. The mechanism responsible for cell death of MCF-7 through autophagy and no apoptosis was not determined.

Freitas et al. (2018) [20] studied *Euterpe oleracea* Mart. seed extract and ethyl acetate fraction and also evidenced that this extract did not cause apoptosis in MCF-7 cell line.

In order to evaluate how açai seed extract induced autophagy, we performed DAF-2A assay. Treatment with 25 μg/mL and 250 μg/mL of açai seed extract caused an increase in ROS production after 6 h when compared to control cells, suggesting the induction of autophagy by ROS increase in açai-treated cells.

Apoptosis and autophagy are cell death programmed mechanisms. Autophagy promotes cell survival by suppressing apoptosis induction, but can also lead to cell death [48].

ROS are associated with many diseases and are key roles in the regulation of apoptosis and autophagy in cancer [49]. Low levels of ROS usually stimulate cell proliferation and survival, whereas excessive levels induce cell death through diminished antioxidant capacity [50].

The role of autophagy in MCF-7 cells is still unclear. One study involved augmenting pro-apoptotic autophagy to induce cell death [51] and the other involves the upregulation of pro-survival drug resistance [52].

The autophagy induction effect of açai can be attributed to its high concentration of polyphenols and flavonoids. Genistein, from *Lupinus luteus*, induced autophagic cell death, and was believed to act through the modulation of antioxidant enzyme and apoptotic signaling pathways in breast cancer cell [53,54]. Rottlerin, a natural polyphenol from *Mallotus philippinensis*, was capable of triggering autophagic cell death in MCF-7 cells by non-canonical signaling cascades [55]. Resveratrol, a polyphenol abundantly found in many plant foods, was reported to inhibit breast cancer stem-like cells growth via the induction of autophagy and the suppression of WNT/b-Catenin signaling pathway [56]

Flavonoids present in many plants are associated with antioxidant activity and can protect cells from oxidative stress. On the other hand, the pro-oxidative effects and cytotoxicity of flavonoids, possibly due to generation of ROS, including H_2_O_2_ and induction of apoptosis have been reported [57,58,59].

Our reports evidenced the presence of: (−)-epicatechin (**1**), cyanidin 3-glucoside (**2**), procyanidin B2 (**3**), kaempferol-3-*O*-rutinoside (**4**), nobiletin (**5**), dihydrokaempferol (**6**), diosmetin (**7**), 3-*O*-Methylquercetin or isorhamnetin (**8**) isoorientin (**9**), genistein-8-C-glucoside (**10**) and apigenin 6,8-digalactoside.

The ability of flavonoids to possess antioxidant activity is dependent on absorption and metabolism of each flavonoid in particular. In general, antioxidant activity of flavonoids is decreased after the metabolic reactions.

Yokomizo and Moriwaki (2006) reported the relation between the uptake of flavonoids and the response of human colon adenocarcinoma Caco-2 cells exposed to oxidative stress. They stated that flavonoid aglycones were incorporated into Caco-2 cells in a concentration- and time-dependent manner, but neither glycosides nor unstable myricetin were incorporated into the cells. The incorporated flavonoids reduced ROS induced by H_2_O_2_ in the cells, but flavonoids with high radical scavenging activity also generated H_2_O_2_ [60]_,_ which could explain our results in which açai seed extract increased ROS production in MCF-7 breast cancer cell line.

Miura et al. also reported that 7 out of 14 flavonoids generated H_2_O_2_ by incubation in acetate buffer, and suggested that pyrogallol or catechol structure was involved in generation [59].

Several recent studies have revealed that anticancer activities of flavonoids may be mediated through pro-oxidant action [61,62] Cancer cells exhibit higher and more persistent oxidative stress and are more susceptible to being killed by drugs that boost increased ROS levels, such as some flavonoids [63,64,65,66] Whether a flavonoid acts as anti- or pro-oxidant depends on its dose, cell type and also culture conditions [65,67,68]

Nitric oxide (NO) is an integral part of ROS produced by nitric oxide synthases [69,70,71]. Excessive ROS may cause irreversible damage to DNA, leading to cell death [72,73]

ROS stimulates autophagy by regulation of ATG4 and stress signaling pathways [74]. Exogenous NO induces autophagy [75,76]. Duan et al. (2014) reported that NO is required for proper autophagic vesicle formation or maturation at a step after LC3 lipidation in breast cancer cells [77]

In conclusion, three distinct phenolic clusters and six phenolic single nodes were identified using MS/MS molecular networking. Twenty chemical compounds were isolated from these clusters and nodes. Nobiletin and genistein-8-c-glucoside were identified for the first time from açai using GNPS platform. Açai seed extract reduced MCF-7 breast cancer cell line viability and induced autophagy and morphologic changes in treated cells. There is a dual effect of flavonoids present in açai. Besides açai’s antioxidant activity, flavonoids with high radical scavenging activity present in açai seed extract have a cytotoxic effect due to generation of ROS.

## Figures and Tables

**Figure 1 molecules-26-03546-f001:**
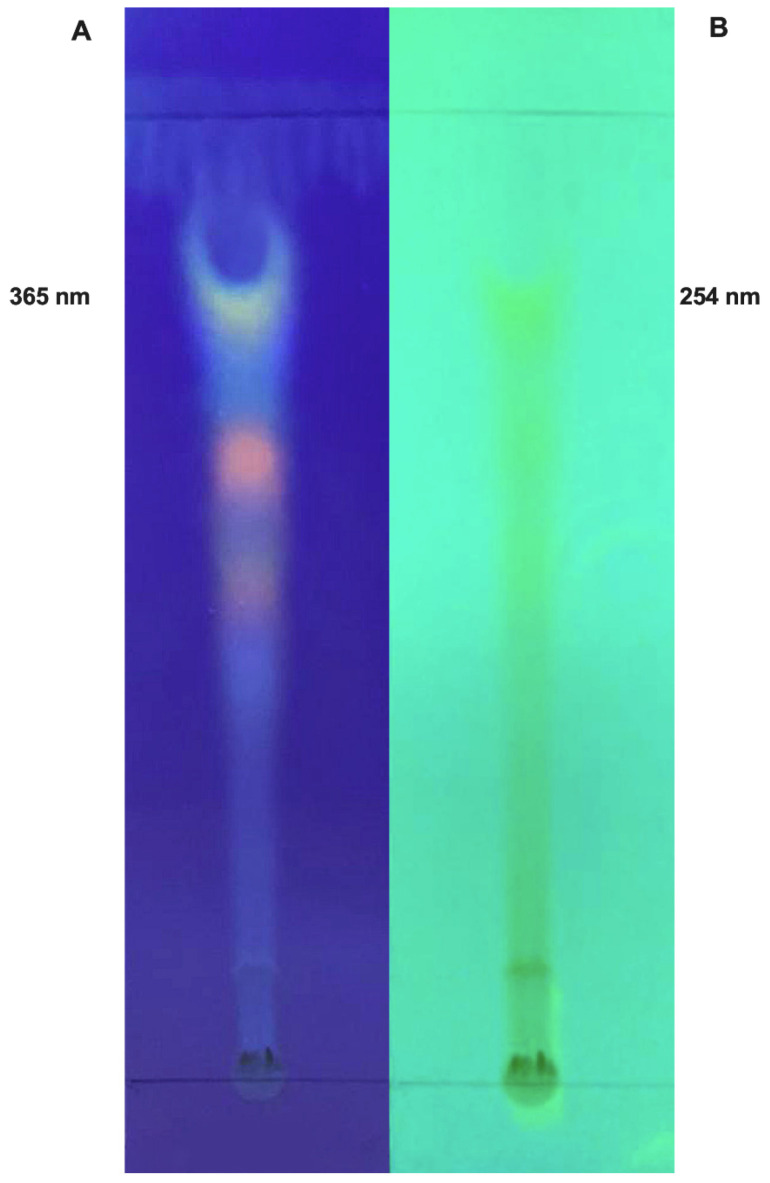
Thin-layer chromatography of hydro-alcoholic açai seed extract at 365 (**A**) and 254 nm (**B**).

**Figure 2 molecules-26-03546-f002:**
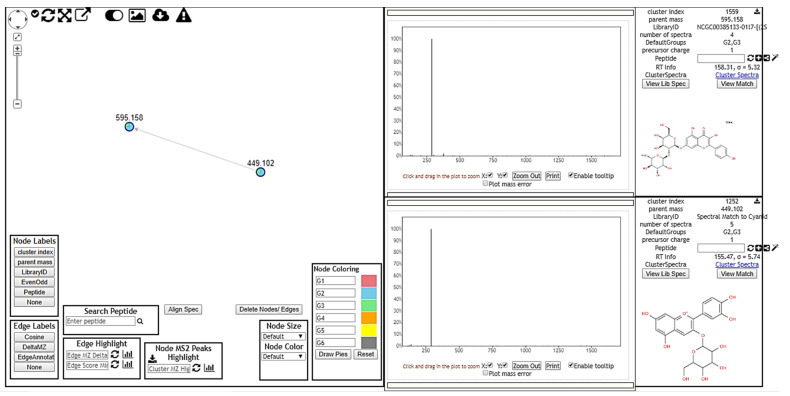
Molecular network creation and visualization. Node labels indicate matches made to GNPS spectral libraries, with additional information displayed with mouseovers. These matches provide users a starting point to annotate unidentified MS/MS spectra within the network. G1, G2 and G3 correspond to seed extract, pulp and total fruit extract, respectively.

**Figure 3 molecules-26-03546-f003:**
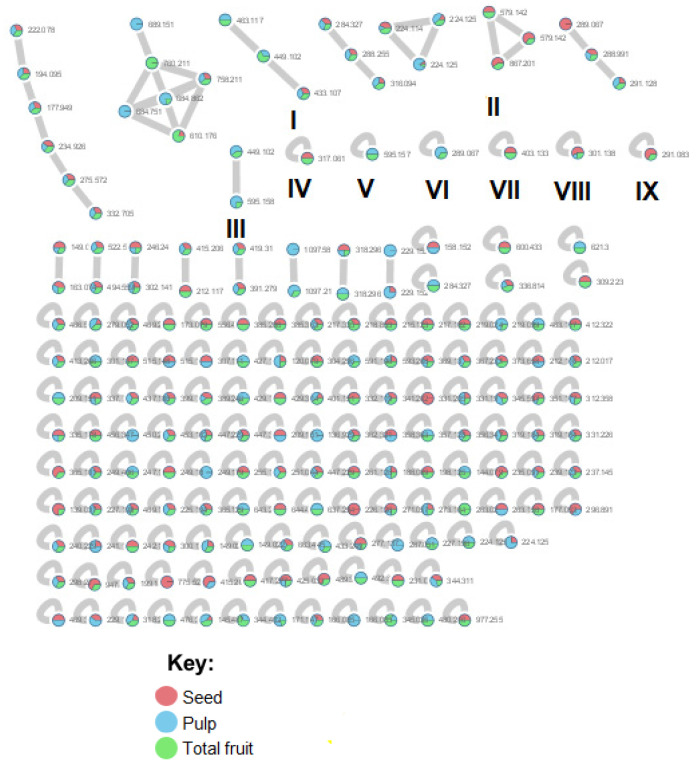
Complete MS/MS network analysis of *Euterpe oleracea* Mart. Compounds with similar fragmentation patterns form clusters in the network (Roman numbers).

**Figure 4 molecules-26-03546-f004:**
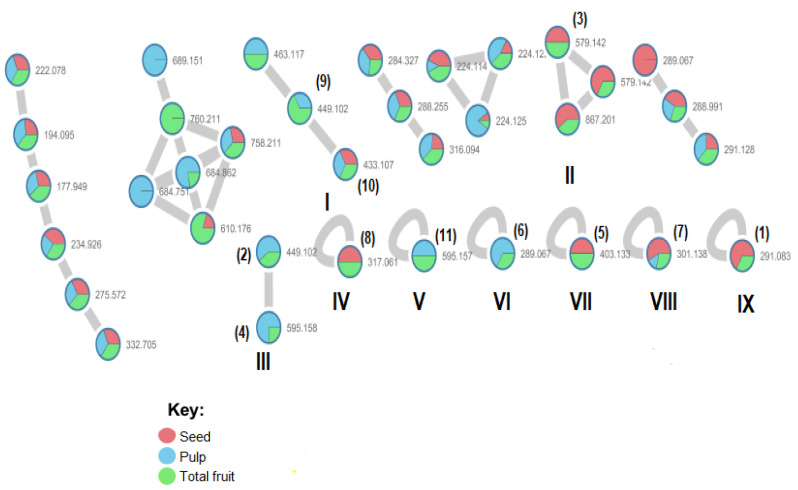
Zoom in of MS/MS network analysis (Figure 2) of seed, pulp and total fruit extracts from *E. oleracea* Mart. Colors indicated in the key correspond to the different extracts (seed, pulp and total fruit). Nodes bolded by blue lines represent the compounds with GNPS hits. Roman numbers represents the clusters and bracketed number represents compounds identified with GNPS hits.

**Figure 5 molecules-26-03546-f005:**
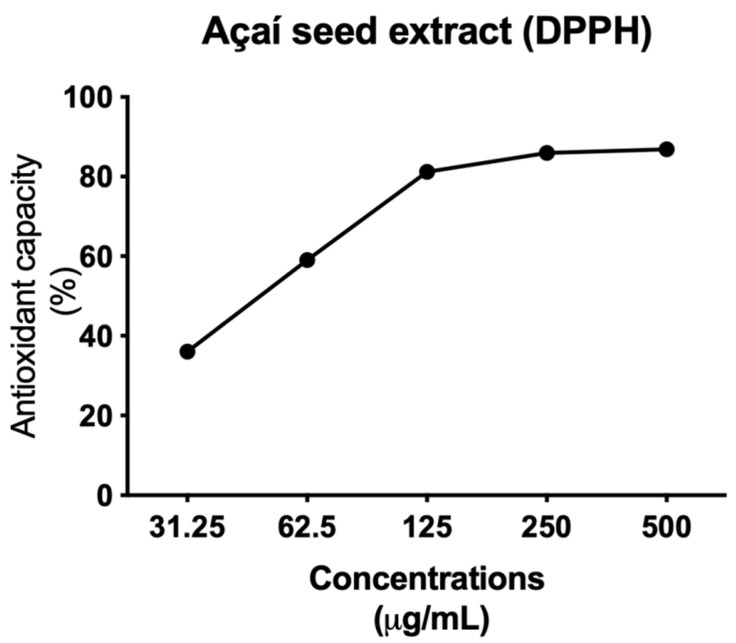
Determination of DPPH radical scavenging activity of açai seed extract. All experiments were performed in triplicate.

**Figure 6 molecules-26-03546-f006:**
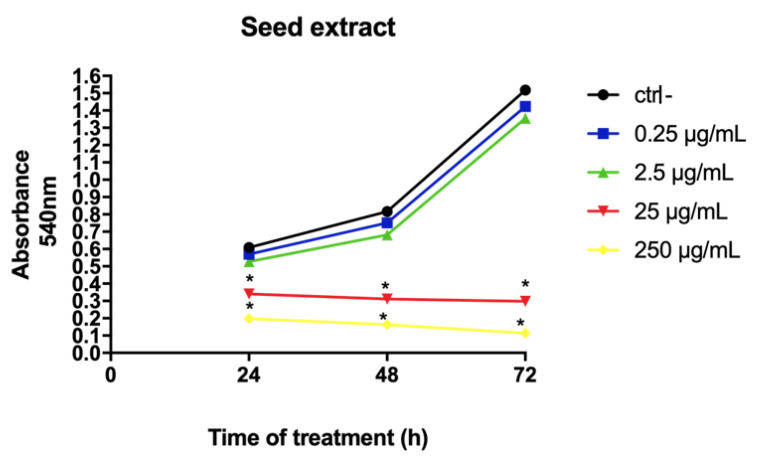
MTT assay analyzing the viability of MCF-7 cells after treatment with the seed extract of açai. (A) The extracts of açai caused significant reduction in cell viability of the treated cells, especially after 72 h of treatment. * *p* < 0.05. One-way analysis of variance (ANOVA) followed by Dunnett’s post hoc test. Ctrl -: cells with DMEM.

**Figure 7 molecules-26-03546-f007:**
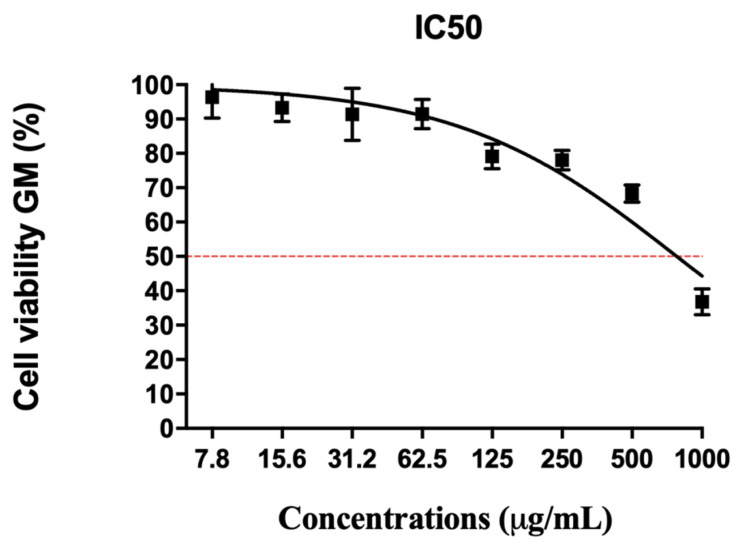
IC 50 curve of açai seed for GM cell viability. Data represent the means ± the SD of viable GM cell percentage in relation to untreated cells after 24 h of treatment.

**Figure 8 molecules-26-03546-f008:**
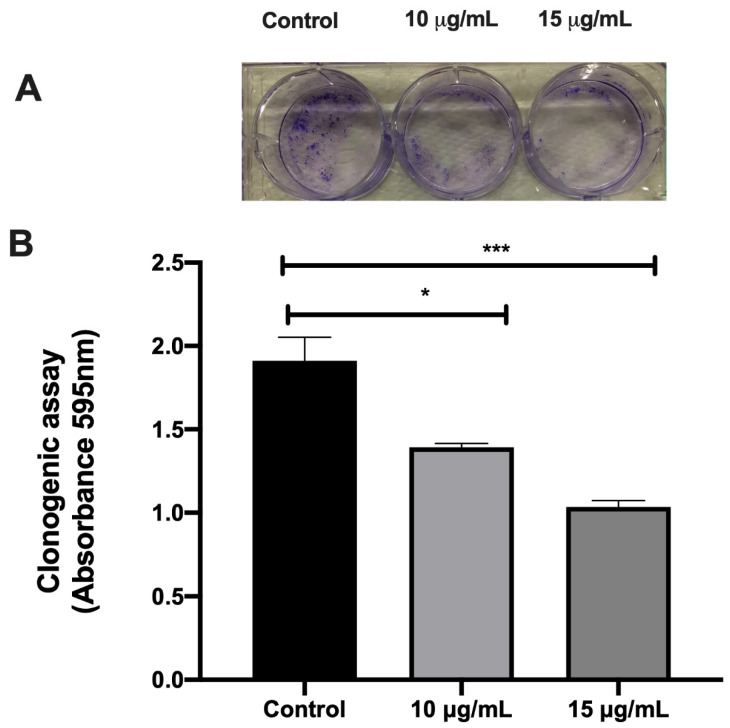
Clonogenicity assay of MCF-7 breast cancer cell line treated with açai seed extract performed after 15 days of culture (**A**). Data were obtained as 595 nm absorbance values (**B**). * *p* < 0.05, *** *p* < 0.001.

**Figure 9 molecules-26-03546-f009:**
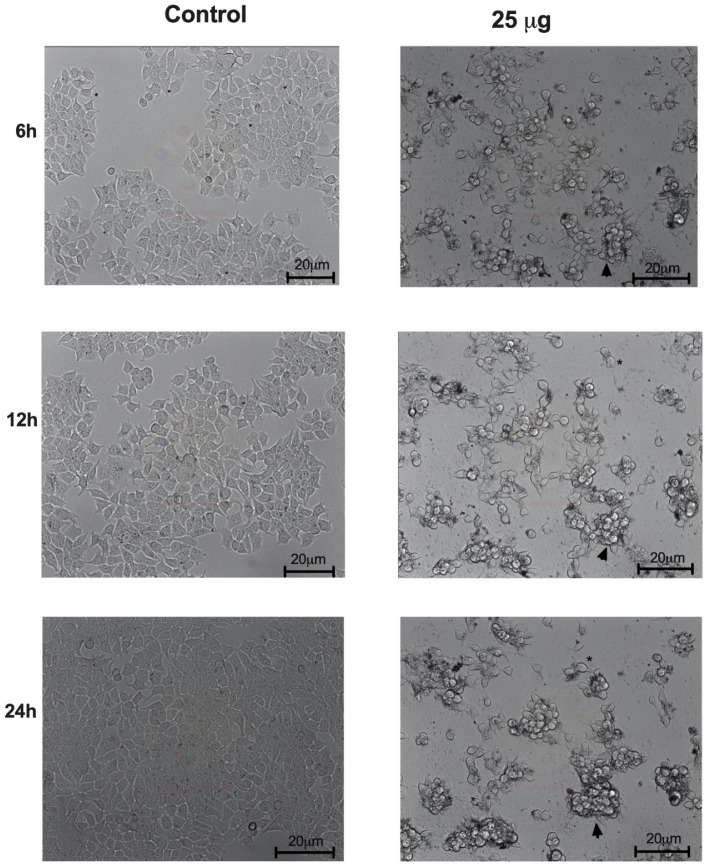
Morphology analysis by time-lapse microscopy of MCF-7 cells treated with açai seed extract (25 μg/mL). Seed extract caused changes in MCF-7 cells, such as cell shrinking, membrane blebbing (arrow) and cell lysis (asterisk).

**Figure 10 molecules-26-03546-f010:**
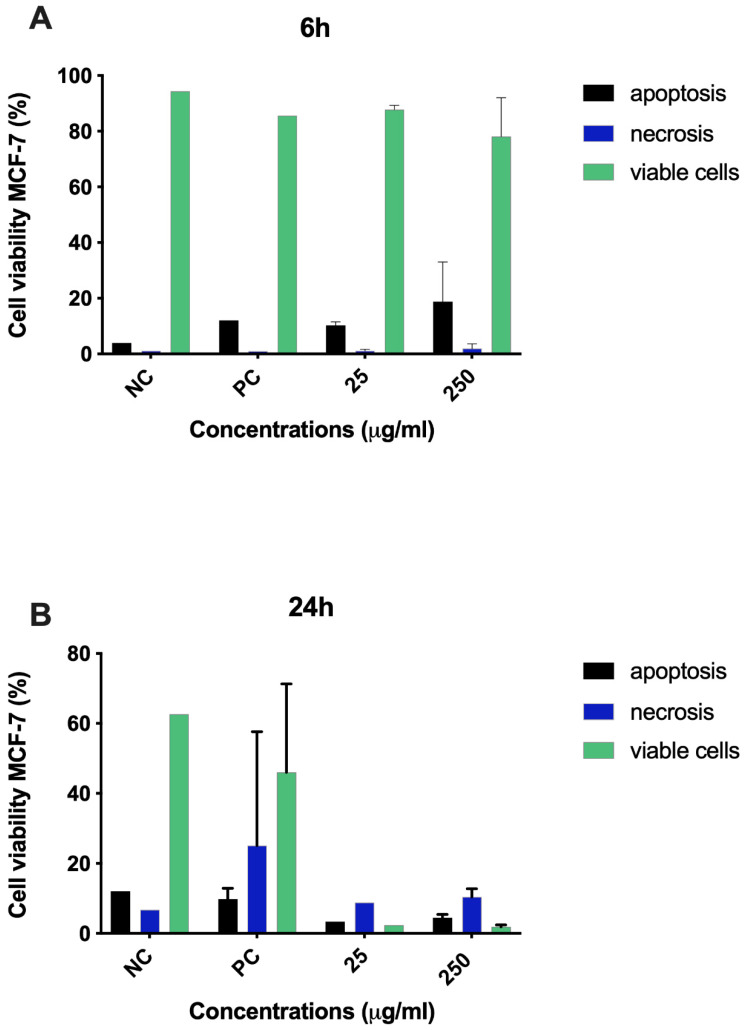
Seed extract did not cause apoptosis on MCF-7 cells. Caspase 3 and 7 assay demonstrated that seed extract did not increase the activity of caspase 3 and caspase 7 when compared to control cells after 6 h (**A**) and 24 h (**B**) of treatment. NC: cells with DMEM; PC: DMSO (100 μL).

**Figure 11 molecules-26-03546-f011:**
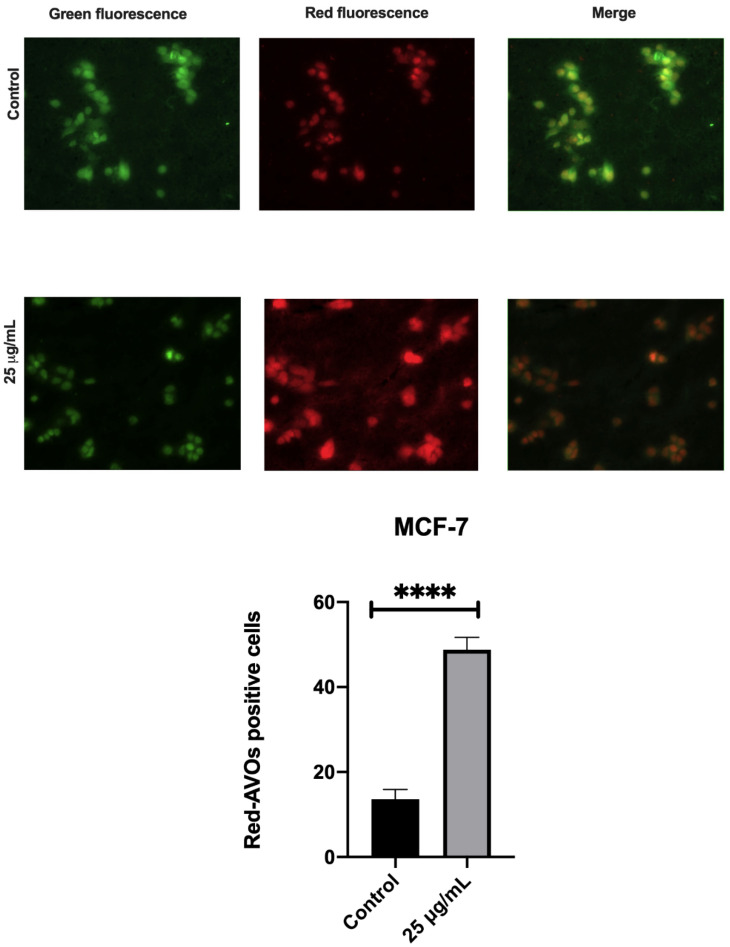
The MCF-7 cells seeded in 3 cm glass bottom plates were treated with 1 μg/mL AO for 45 min and then images were recorded under fluorescence microscopy. Acridine-orange (AO) stained red fluorescent-positive acidic vesicular organelles (AVOs) were increased in açai seed extract-treated cells. Red fluorescence indicated the AO-stained AVOs within the cells. Representative results from three independent experiments with similar results are shown. The percentage of red-AVO positive cells was calculated in 5 photos of each condition and the results are presented in the lower histogram. Representative blots and photos from three independent experiments with similar results are shown. **** *p* < 0.0001. Unpaired T test (scale bars, 50 μm).

**Figure 12 molecules-26-03546-f012:**
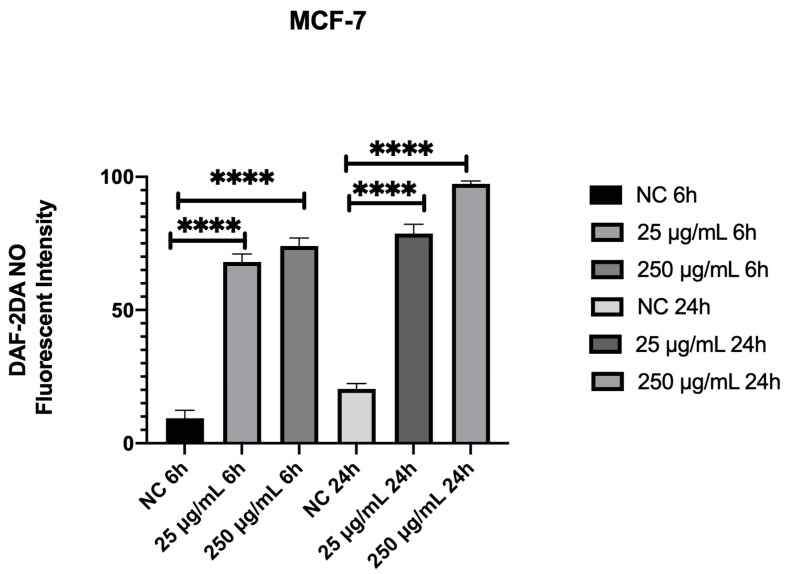
Effect of the extract on the NO levels expressed as percentage difference of fluorescence in MCF-7 breast cancer cell line(~1 × 10^5^ cells/mL) using DAF-2DA (10 µM) incubated for 30 min. Representative data from experiments in triplicate. Basal production was considered zero and the difference in fluorescence was expressed in the compound’s incubation times (6 h, 24 h). Values expressed as mean ± s.p.m. ANOVA **** *p* < 0.0001. NC: negative control.

**Table 1 molecules-26-03546-t001:** Identification of chemical compound extracts from açai by LC-ESI-MS.

Compound	Structure	Calculated *m/z*	Observed *m/z*	Error (ppm)	Parts from Açai
1	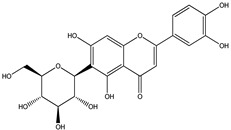	291.0863	291.0848	−5.15	seed and total fruit
2	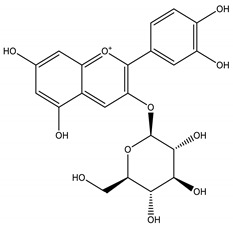	449.1078	449.1056	−4.89	pulp and total fruit
3	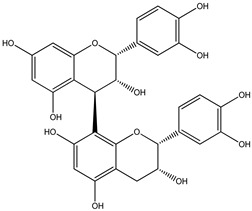	579.1497	579.1481	−2.76	seed and total fruit
4	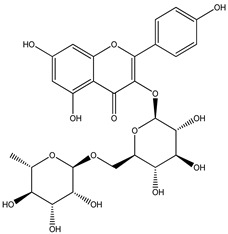	595.1657	595.1628	−4.87	pulp and total fruit
5	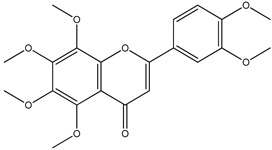	403.1387	403.1371	−3.96	seed and total fruit
6	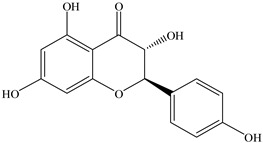	289.0707	289.0695	−4.15	pulp and total fruit
7	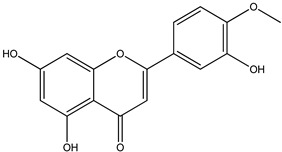	301.0707	301.0695	−3.98	seed and total fruit
8	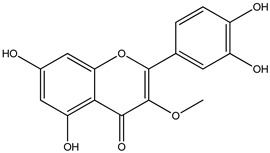	317.0656	317.0643	−4.10	seed and total fruit
9	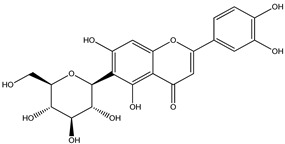	449.1078	449.1060	−4.00	pulp and total fruit
10	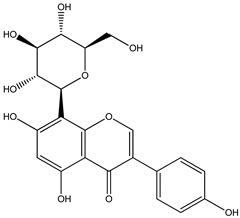	433.1129	433.1110	−4.38	seed, pulp and total fruit
11	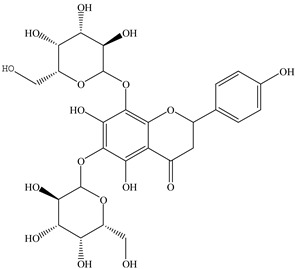	595.1657	595.1632	−4.20	pulp and total fruit

## Data Availability

The data and materials are available with the researchers at Faculty of Medical Science, UNICAMP. All datasets presented in this study are included in the article.

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
