# Peer review of "Açai (Euterpe oleracea Mart.) Seed Extract Induces ROS Production and Cell Death in MCF-7 Breast Cancer Cell Line"

_molecules, 2021, doi:10.3390/molecules26123546_

Round 1

Reviewer 1 Report

The manuscript describes the use of a digital platform to identify MS / MS results in order to elucidate the chemical composition of acai seed extracts (as well as pulp and whole fruit). In addition, the authors investigate the bioactivities of açaí seed extracts. Overall, the results are interesting for the scientific community, however the article should have been read and revised more carefully by the authors before its submission. Authors should not submit manuscripts that have not been previously reviewed in a coherent and objective manner. For example, the introduction contains several scientific errors; the description of the methodology, which is so easy for those who do the experimental work, was done without care and is full of errors that have to be corrected. Given the relevance of the content of the experimental results, I am in favour of the publication, however an extensive revision is necessary.

Next, I explain sequentially the points in the manuscripts that require correction / revision.

Authors should define all non-common abbreviations at first mention, e.g. GNPS in abstract.

Please note that the word açaí, is being written in three different ways, with and without acute accent (açaí and açai) and as “acai”, please correct and standardize the whole document.

Line 85, only one cell line is used, replace “lines” by “line”. Correct in whole document.

Line 95, write the species descriptor correctly (Mart.). In addition, the descriptor must be placed after the name of the species in the first mention, there is no need to write it each time the species is referred along the manuscript. Also, verify that the way it is written (species and descriptor) varies throughout the document (e.g. line 2, 117, 130 ...).

In line 65, the authors present the nutritional composition of açaí, but do not refer to carbohydrates, sugars and fibres. In my opinion, authors could present more accurate data, with % of each component, in order to give a realistic idea about the main nutritional compounds.

Line 67 to 68, please revise. This sentence is not scientifically correct. Please verify if all listed compounds are in fact phytochemicals, and if have antioxidant activity. For example, starch and fibers are not phytochemicals.

Line 77-78, please rephrase, the sentence is ambiguous.

Line 158-159, please correct the decimal separator (a dot should be used instead of a comma).

Section 2.6 the authors mention that culture media was deprived of antibiotics, why? Did you add L-glutamine, or not? If not, why? If yes, please complete the methods.

Authors must be consistent in the system of units and in the way of representing them, authors must use the same notation throughout the manuscript. Examples of different notations for the same units: section 2.3 litre with a capital L, section 2.6, litre with a lowercase l;

Section 2.7. Line 162. Please add the density of GM cells.

For a better understanding authors could join sections 2.6 and 2.7, and describe the method sequentially in time, as the density of seeding cells in 96-well plates should come first than cell treatment… please revise.

In section 2.7, For cell viability assay, did authors remove the supernatant from cells and then added 10 microliters of MTT per well? But, 10 uL does not cover the bottom of the well... Is this correct? Also, after adding MTT, where were placed the cells and at which temperature? Which solution was used for cell wash, and explain why performed this step. And, add the volume of DMSO used per well. Please revise this section accordingly.

Please explain in which solvent extracts were dissolved, and which was the concentration of mother solutions (stock solutions).

Line 172, add the ethanol concentration.

Line 173-175. Authors should detail the method. As in Figure 8, photos are shown, authors should present the number of colonies formed under each condition.

Line 180. “Cells were treated with 25 and 250μg/ml for 6 and 24h”, of what? Please complete.

Section 2.7. Which was the probe used to detect ROS? DAF-2DA diacetate, mentioned in the section title (line 192) and DCF-DA (line 197) are two different probes. Please clarify.

Did authors lyse the cells? (line 196) If cells are lysed what was analysed by flow cytometry?

However, as shown in Figure 12, it seems that authors measured NO, not ROS, thus I presume DAF-2DA diacetate was used.

Why authors prefer to measure NO rather than total ROS? Why NO is relevant in this cell line? Please explain.

Line 217. Ref 39, is not adequate to describe the behaviour of acridine orange, since this article does not mention this dye. However, the sentence (Lines 215-217) was extracted without modification from 10.1038/s41598-017-13904-0 (not referred in manuscript). Correct references.

Figure 1. Authors should mark the TLC images (with A and B, for example) and make the correct correspondence to the wavelength. Complete legend.

Why the authors also did not do TLC for the pulp and complete extracts.?

Figure 3. The quality of figure is very low. Please increase the quality of final figure. To a better understanding authors should explain figure details in its legend. Indicate the meaning of roman numbers…

Figure 4 legend. What do authors mean with “different growth media source”?

Please be more concise and precise in describing this figure.

Figure 5. Please correct the xx axis legend, the decimal separator is a dot.

Figure 6. Correct the decimal separators if figure legend.

Figure 7. Please indicate in legend the period of treatment. Figure resolution must be increased. This is a curve of cell viability with and intercept at 50% viability.

Figure 8. Panel A. Authors should identify each of the 3 wells (what represents the left well, for example? This must be indicated).

Based on results obtained in figure 6, authors should explain the rational to choose the concentrations used to perform colonogenic assay.

Figure 10 legend. Authors should mention what is the positive and the negative control.

Line 420, replace genistein-8-c-glucoside by genistein-8-C-glucoside

The increase in NO might not induce oxidative stress in MCF-7 cells as it diffuses out of the cell. Why measure NO instead of total ROS (using DCFDA)? Some authors have reported that activation of MCF-7 nitric oxide synthase, producing NO that is released from the cell is involved in the activation of vascular cells growth and in the promotion of tumorigenesis (e.g. DOI: 10.4161/cbt.6.7.4330; DOI: 10.1158/0008-5472.CAN-05-4623). Here authors report cell death and autophagy. How do you correlate your results with the activation of nitric oxide synthase?  

Authors could enrich the discussion by correlating results from different methodologies.

Please revise the whole manuscript concerning grammar and spelling (verb tenses, agreement between verb and number (singular and plural)).

Also make sure that the text in introduction is not exactly the same as that in consulted reference.

Author Response

Response to reviewer 1

We would like to thank you for you reviewer report. In order to your requests, we updated the whole manuscript.

The manuscript describes the use of a digital platform to identify MS/MS results in order to elucidate the chemical composition of acai seed extracts (as well as pulp and whole fruit). In addition, the authors investigate the bioactivities of açaí seed extracts. Overall, the results are interesting for the scientific community, however the article should have been read and revised more carefully by the authors before its submission. Authors should not submit manuscripts that have not been previously reviewed in a coherent and objective manner. For example, the introduction contains several scientific errors; the description of the methodology, which is so easy for those who do the experimental work, was done without care and is full of errors that have to be corrected. Given the relevance of the content of the experimental results, I am in favour of the publication, however an extensive revision is necessary.

Next, I explain sequentially the points in the manuscripts that require correction / revision.

Authors should define all non-common abbreviations at first mention, e.g. GNPS in abstract.

Response: The abbreviations were defined

Please note that the word açaí, is being written in three different ways, with and without acute accent (açaí and açai) and as “acai”, please correct and standardize the whole document.

 Response: The word açai was defined in all article

Line 85, only one cell line is used, replace “lines” by “line”. Correct in whole document.

Response: Corrected

Line 95, write the species descriptor correctly (Mart.). In addition, the descriptor must be placed after the name of the species in the first mention, there is no need to write it each time the species is referred along the manuscript. Also, verify that the way it is written (species and descriptor) varies throughout the document (e.g. line 2, 117, 130 ...). 

Response: Corrected

In line 65, the authors present the nutritional composition of açaí, but do not refer to carbohydrates, sugars and fibres. In my opinion, authors could present more accurate data, with % of each component, in order to give a realistic idea about the main nutritional compounds.

Response: We included the percentage of the nutritional composition of açai.

Line 67 to 68, please revise. This sentence is not scientifically correct. Please verify if all listed compounds are in fact phytochemicals, and if have antioxidant activity. For example, starch and fibers are not phytochemicals.

Response: We have rewritten the sentence.

Line 77-78, please rephrase, the sentence is ambiguous.

Response: We have rewritten the sentence.

Line 158-159, please correct the decimal separator (a dot should be used instead of a comma).

 Response: Corrected

Section 2.6 the authors mention that culture media was deprived of antibiotics, why? Did you add L-glutamine, or not? If not, why? If yes, please complete the methods.

 Response: We corrected the methods about culture media, including the antibiotics. L-glutamine was not used because lab protocols.

Authors must be consistent in the system of units and in the way of representing them, authors must use the same notation throughout the manuscript. Examples of different notations for the same units: section 2.3 litre with a capital L, section 2.6, litre with a lowercase l;

 Response: Corrected

Section 2.7. Line 162. Please add the density of GM cells.

Response: We included the density of GM cells.

For a better understanding authors could join sections 2.6 and 2.7, and describe the method sequentially in time, as the density of seeding cells in 96-well plates should come first than cell treatment… please revise.

 Response: We have rewritten these sections, joining 2.6 and 2.7 section

In section 2.7, For cell viability assay, did authors remove the supernatant from cells and then added 10 microliters of MTT per well? But, 10 uL does not cover the bottom of the well... Is this correct? Also, after adding MTT, where were placed the cells and at which temperature? Which solution was used for cell wash, and explain why performed this step. And, add the volume of DMSO used per well. Please revise this section accordingly.

 Response: We corrected this information (100 uL). The cells were placed at 37°C and incubated. Cells were washed in PBS 3 times and 100 uL of DMSO was used.

Please explain in which solvent extracts were dissolved, and which was the concentration of mother solutions (stock solutions).

 Response: The samples were diluted in stock solutions in dimethyl sulfoxide (DMSO) ® (Merck) at a concentration of 0.1 g/mL.

Line 172, add the ethanol concentration.

 Response: The ethanol concentration was 20%.

Line 173-175. Authors should detail the method. As in Figure 8, photos are shown, authors should present the number of colonies formed under each condition.

  Response: We have better described the methods of clonogenic assay.

Line 180. “Cells were treated with 25 and 250μg/ml for 6 and 24h”, of what? Please complete.

  Response: We included açai seed extract.

Section 2.7. Which was the probe used to detect ROS? DAF-2DA diacetate, mentioned in the section title (line 192) and DCF-DA (line 197) are two different probes. Please clarify.

 Response: We used DAF-2DA for NO detection.

Did authors lyse the cells? (line 196) If cells are lysed what was analysed by flow cytometry?

 Response: The cells were not lysed. We corrected this information.

However, as shown in Figure 12, it seems that authors measured NO, not ROS, thus I presume DAF-2DA diacetate was used.

Why authors prefer to measure NO rather than total ROS? Why NO is relevant in this cell line? Please explain.

 Response: Nitric oxide (NO) is an integral part of ROS produced by nitric oxide synthases. Excessive ROS may cause irreversible damage to DNA leading to cell death

ROS stimulates autophagy by regulation of ATG4 and stress signaling pathways. Exogenous NO induces autophagy. Duan et al (2014) reported that NO is required for proper autophagic vesicle formation or maturation at a step after LC3 lipidation in breast cancer cells.

Line 217. Ref 39, is not adequate to describe the behaviour of acridine orange, since this article does not mention this dye. However, the sentence (Lines 215-217) was extracted without modification from 10.1038/s41598-017-13904-0 (not referred in manuscript). Correct references.

  Response: We replaced this reference and corrected the sentence.

Figure 1. Authors should mark the TLC images (with A and B, for example) and make the correct correspondence to the wavelength. Complete legend.

 Response: Corrected

Why the authors also did not do TLC for the pulp and complete extracts.?

  Response: Further experiments were performed only for seed extract because we have another manuscripts regarding biological effects of açai seed extract in breast cancer cell line

Figure 3. The quality of figure is very low. Please increase the quality of final figure. To a better understanding authors should explain figure details in its legend. Indicate the meaning of roman numbers…

  Response: We corrected this.

Figure 4 legend. What do authors mean with “different growth media source”?

Please be more concise and precise in describing this figure.

  Response: We better described the legend.

Figure 5. Please correct the xx axis legend, the decimal separator is a dot.

  Response: Corrected

Figure 6. Correct the decimal separators if figure legend.

  Response: Corrected.

Figure 7. Please indicate in legend the period of treatment. Figure resolution must be increased. This is a curve of cell viability with and intercept at 50% viability.

 Response: We indicated the period of treatment and increased figure resolution.

Figure 8. Panel A. Authors should identify each of the 3 wells (what represents the left well, for example? This must be indicated).

 Response: We have included the wells identification.

Based on results obtained in figure 6, authors should explain the rational to choose the concentrations used to perform clonogenic assay.

  Response: 25 ug/mL was too cytotoxic to cells, so we choose an intermediate dose to perform clonogenic assay

Figure 10 legend. Authors should mention what is the positive and the negative control.

  Response: We included positive and negative information.

Line 420, replace genistein-8-c-glucoside by genistein-8-C-glucoside

  Response: Replaced

The increase in NO might not induce oxidative stress in MCF-7 cells as it diffuses out of the cell. Why measure NO instead of total ROS (using DCFDA)? Some authors have reported that activation of MCF-7 nitric oxide synthase, producing NO that is released from the cell is involved in the activation of vascular cells growth and in the promotion of tumorigenesis (e.g. DOI: 10.4161/cbt.6.7.4330; DOI: 10.1158/0008-5472.CAN-05-4623). Here authors report cell death and autophagy. How do you correlate your results with the activation of nitric oxide synthase?  

  Response: Nitric oxide (NO) is an integral part of ROS produced by nitric oxide synthases. Excessive ROS may cause irreversible damage to DNA leading to cell death

ROS stimulates autophagy by regulation of ATG4 and stress signaling pathways. Exogenous NO induces autophagy. Duan et al (2014) reported that NO is required for proper autophagic vesicle formation or maturation at a step after LC3 lipidation in breast cancer cells.

Authors could enrich the discussion by correlating results from different methodologies.

Response: We have improved the discussion.

Please revise the whole manuscript concerning grammar and spelling (verb tenses, agreement between verb and number (singular and plural)).

Also make sure that the text in introduction is not exactly the same as that in consulted reference.

Response: Corrected.

Reviewer 2 Report

The article is well written and provides valuable data in the field. 

However, I would have the following minor remarks:

In Fig 6. it is not clear which values are compared with each other (as shown in Fig 8 for example);

In Figs 5 and 10. the data should be compared statistically and commented accordingly. 

Author Response

Response to reviewer 2

The article is well written and provides valuable data in the field. 

However, I would have the following minor remarks:

In Fig 6. it is not clear which values are compared with each other (as shown in Fig 8 for example);

Response: The concentrations tested were compared with control (cells with DMEM)

In Figs 5 and 10. the data should be compared statistically and commented accordingly. 

Response: Corrected

Reviewer 3 Report

The manuscript attempts to present the results of extraction compounds from Acai seed, pulp and complete fruit, the analysis of MS spectra by using Global Natural Products Social Molecular Networking (GNPS), an open-access knowledge base for identification of natural products within, and determination of some biological activities of acai seed extracts.

However, I cannot recommend this manuscript for publishing in Molecules because results are not presented in clear, unambiguous and/or proper way and there are serious flaws.

Here only some of them are listed:

It is common to compare results of in vitro assays for novel samples with those of referent compounds. Such referent data and comparison are missing or are not properly described in this study. For example, for MTT and DPPH tests no results for referent compounds are shown, while in Figure 10 it is not clear which compounds are NC and PC.

It is not clear does the extraction procedure described in lines 101-106 refer to all three types of samples or only to seed.

It is not clearly described why the biological testing was performed only on seed extract.

It is not clear presented and discussed the author's statement that Acai seed extract induces generation of hydrogen peroxide.

It is not clear does compounds listed in Table 1 were identified by using GNPS since Figures 3 and 4 are of very low quality. Numeration of compounds within Table 1 does not correspond to those within the text! In Table 1 names of NP aglycons should be given. The GNPS usage for dereplication and assignation is not presented sufficiently to be credible.

At first, English should be corrected. For example (one of several), in lines 158- 159 „were treated with 0,25, 2,5, 25 and 158 250μg/ml OF WHAT? for 24, 48 and 72h. GM cultured cells were treated with 7,8 to 1000 μg/ml OF WHAT? for 24 h.“ Abbreviations should be defined eg GM , particularly in Abstract if they are not standard ones (e.g. GNPS). In Abstract, there is repetition and statements which have not been shown in the study (H2O2).

Figures 2 – 4 are very confusing and in figure captions it is not described enough what they present and how they should be interpreted.

In Figure 10, it is not clear what is on y-axis based on the Figure caption.

Author Response

Response to reviewer 3

The manuscript attempts to present the results of extraction compounds from Acai seed, pulp and complete fruit, the analysis of MS spectra by using Global Natural Products Social Molecular Networking (GNPS), an open-access knowledge base for identification of natural products within, and determination of some biological activities of acai seed extracts.

However, I cannot recommend this manuscript for publishing in Molecules because results are not presented in clear, unambiguous and/or proper way and there are serious flaws.

Here only some of them are listed:

It is common to compare results of in vitro assays for novel samples with those of referent compounds. Such referent data and comparison are missing or are not properly described in this study. For example, for MTT and DPPH tests no results for referent compounds are shown, while in Figure 10 it is not clear which compounds are NC and PC.

Response: The experiments were performed with açai seed extract, not with isolated compounds. NC: negative control; PC: positive control.

It is not clear does the extraction procedure described in lines 101-106 refer to all three types of samples or only to seed.

Response: After thawing at room temperature, the sample was separated into three parts: seed, pulp and total fruit (seed + pulp). The extraction process followed the methodology developed by Moura et al (2012).

It is not clearly described why the biological testing was performed only on seed extract.

Response: The biological tests, excepted MS/MS analysis, were performed only on seed extract. In methodology sections, we described that 0.25, 2.5, 25 and 250ug/mL of açai seed extract were used for biological tests.

It is not clear presented and discussed the author's statement that Acai seed extract induces generation of hydrogen peroxide.

Response: We better described the results and methodology.

It is not clear does compounds listed in Table 1 were identified by using GNPS since Figures 3 and 4 are of very low quality. Numeration of compounds within Table 1 does not correspond to those within the text! In Table 1 names of NP aglycons should be given. The GNPS usage for dereplication and assignation is not presented sufficiently to be credible.

Response: The compounds in Table 1 are exactly as described in the text. The figures quality was improved.

At first, English should be corrected. For example (one of several), in lines 158- 159 „were treated with 0,25, 2,5, 25 and 158 250μg/ml OF WHAT? for 24, 48 and 72h. GM cultured cells were treated with 7,8 to 1000 μg/ml OF WHAT? for 24 h.“ Abbreviations should be defined eg GM , particularly in Abstract if they are not standard ones (e.g. GNPS). In Abstract, there is repetition and statements which have not been shown in the study (H2O2).

Response: We corrected this information.

Figures 2 – 4 are very confusing and in figure captions it is not described enough what they present and how they should be interpreted.

Response: Corrected

In Figure 10, it is not clear what is on y-axis based on the Figure caption.

Response: Corrected

Round 2

Reviewer 1 Report

The authors have addressed the questions and corrected/clarified the manuscript. The figures quality was clearly improved. The manuscript is ready to be published after some minor corrections,

Line 68, please replace 9,1% by 9.1%.

Line 161, please replace “penicilli” by penicillin

in order to have a single writing style of units, I request that the authors standardize the units throughout the manuscript. For example, separate the numbers from the units along the whole manuscript (some are separated others are not), see as example: line 161 (replace 60mg/L by 60 mg/L); do not leave spaces such as, line 163: (replace 0.1 g / mL by 0.1 g/mL). standardize along the document.

Author Response

The authors have addressed the questions and corrected/clarified the manuscript. The figures quality was clearly improved. The manuscript is ready to be published after some minor corrections,

Line 68, please replace 9,1% by 9.1%.

Response: Corrected

Line 161, please replace “penicilli” by penicillin

Response: Replaced

in order to have a single writing style of units, I request that the authors standardize the units throughout the manuscript. For example, separate the numbers from the units along the whole manuscript (some are separated others are not), see as example: line 161 (replace 60mg/L by 60 mg/L); do not leave spaces such as, line 163: (replace 0.1 g / mL by 0.1 g/mL). standardize along the document.

Response: We have corrected this in whole manuscript.

Reviewer 3 Report

I cannot recommend this manuscript for publishing in Molecules because results are still presented in unclear, ambiguous and/or improper way and there are serious flaws such as not definition of control experiments.

Here are comments on what still it not clear enough:

R3 It is common to compare results of in vitro assays for novel samples with those of referent compounds. Such referent data and comparison are missing or are not properly described in this study. For example, for MTT and DPPH tests no results for referent compounds are shown, while in Figure 10 it is not clear which compounds are NC and PC.

Response: The experiments were performed with açai seed extract, not with isolated compounds. NC: negative control; PC: positive control.

R3 comment to Response The response is NOT specific enough. There is no clearly defined which compounds/samples were used as PC and NC in DPPH and MTT tests. For DPPH test, no data for PC and NC are provided. What was PC for DPPH, for example? Vitamin C or?

R3 It is not clear does the extraction procedure described in lines 101-106 refer to all three types of samples or only to seed.

Response: After thawing at room temperature, the sample was separated into three parts: seed, pulp and total fruit (seed + pulp). The extraction process followed the methodology developed by Moura et al (2012).

R3 comment to Response It is still NOT specified enough in the line 103 on what sample(s) “Approximately 360 g of açaí (what?) …” …particularly since in the line 112 it is referred to “hydroalcoholic açaí seed extract”.

R3 It is not clearly described why the biological testing was performed only on seed extract.

Response: The biological tests, excepted MS/MS analysis, were performed only on seed extract. In methodology sections, we described that 0.25, 2.5, 25 and 250ug/mL of açai seed extract were used for biological tests.

R3 comment to Response The authors did NOT answer on the question why was the biological testing performed only on seed extract

It is not clear presented and discussed the author's statement that Acai seed extract induces generation of hydrogen peroxide.

Response: We better described the results and methodology.

R3 comment to Response It should be denoted in which lines of the revised text it is described now

It is not clear does compounds listed in Table 1 were identified by using GNPS since Figures 3 and 4 are of very low quality. Numeration of compounds within Table 1 does not correspond to those within the text! In Table 1 names of NP aglycons should be given. The GNPS usage for dereplication and assignation is not presented sufficiently to be credible.

Response: The compounds in Table 1 are exactly as described in the text. The figures quality was improved.

R3 comment to Response In lines 300-303 “The major compounds were: (-)-epicatechin (1), cyanidin 3-glucoside (2), procyanidin B2 (3), kaempferol-3-O-rutinoside (4), nobiletin (5), dihydrokaempferol (6), dios-301 metin (7), 3-O-Methylquercetin or isorhamnetin (8) isoorientin (9), genistein-8-C-glucoside (10) and 302 apigenin 6,8-digalactoside (11).” In Table 1, for example the compound 1 is a derivate of luteolin, derivative of (-)-epicatechin is a compound 3 in Table 1 and there is no (-)-epicatechin in Table1 and so on. The GNPS usage for dereplication and assignation is not presented sufficiently to be credible.

R3 At first, English should be corrected. For example (one of several), in lines 158- 159 „were treated with 0,25, 2,5, 25 and 158 250μg/ml OF WHAT? for 24, 48 and 72h. GM cultured cells were treated with 7,8 to 1000 μg/ml OF WHAT? for 24 h.“ Abbreviations should be defined eg GM , particularly in Abstract if they are not standard ones (e.g. GNPS). In Abstract, there is repetition and statements which have not been shown in the study (H2O2).

Response: We corrected this information.

R3  Figures 2 – 4 are very confusing and in figure captions it is not described enough what they present and how they should be interpreted.

Response: Corrected

R3 comment to Response Not enough to enable interpretation of the data obtained by GNPS. The authors do not refer within Results and Discussion to the figures 2-4.

In Figure 10, it is not clear what is on y-axis based on the Figure caption.

Response: Corrected

R3 comment to Response

In Figure 10 , by, for example, black colour they are marked percentage of cells which went into apoptosis after each of 4 different treatment or?  It is also not clear sufficiently what were NC and PC and in which concentration.

In each of Figure captions it should be described what were PC and NC if there were used in the experiment.

Author Response

I cannot recommend this manuscript for publishing in Molecules because results are still presented in unclear, ambiguous and/or improper way and there are serious flaws such as not definition of control experiments.

Here are comments on what still it not clear enough:

R3 It is common to compare results of in vitro assays for novel samples with those of referent compounds. Such referent data and comparison are missing or are not properly described in this study. For example, for MTT and DPPH tests no results for referent compounds are shown, while in Figure 10 it is not clear which compounds are NC and PC.

Response: The experiments were performed with açai seed extract, not with isolated compounds. NC: negative control; PC: positive control.

R3 comment to Response The response is NOT specific enough. There is no clearly defined which compounds/samples were used as PC and NC in DPPH and MTT tests. For DPPH test, no data for PC and NC are provided. What was PC for DPPH, for example? Vitamin C or?

Response: For MTT assay, Six wells were included for control (DMEM). Positive control: cells treated with DMSO. For DPPH, A blank sample was prepared using ethanol instead of extract. Trolox was used as positive control.

R3 It is not clear does the extraction procedure described in lines 101-106 refer to all three types of samples or only to seed.

Response: After thawing at room temperature, the sample was separated into three parts: seed, pulp and total fruit (seed + pulp). The extraction process followed the methodology developed by Moura et al (2012).

R3 comment to Response It is still NOT specified enough in the line 103 on what sample(s) “Approximately 360 g of açaí (what?) …” …particularly since in the line 112 it is referred to “hydroalcoholic açaí seed extract”.

Response: Approximately 360 g of of each different portion of açai (seed, pulp and total fruit) were washed in tap water and boiled in distilled water for 5 to 10 minutes.

R3 It is not clearly described why the biological testing was performed only on seed extract.

Response: The biological tests, excepted MS/MS analysis, were performed only on seed extract. In methodology sections, we described that 0.25, 2.5, 25 and 250ug/mL of açai seed extract were used for biological tests.

R3 comment to Response The authors did NOT answer on the question why was the biological testing performed only on seed extract

Response: Our research group initially studied açaí extract (seed, pulp and total fruit) in different cell lines (MCF-7, CACO-2, HT-29, MDA-MB 468)  (doi: 10.1186/1472-6882-14-175). The best results were provided with açaí seed extract in MCF-7 cel line. Because of that, we performed biological tests only with seed extract.

It is not clear presented and discussed the author's statement that Acai seed extract induces generation of hydrogen peroxide.

Response: We better described the results and methodology.

R3 comment to Response It should be denoted in which lines of the revised text it is described now

Response: Lines 1047-1052.

It is not clear does compounds listed in Table 1 were identified by using GNPS since Figures 3 and 4 are of very low quality. Numeration of compounds within Table 1 does not correspond to those within the text! In Table 1 names of NP aglycons should be given. The GNPS usage for dereplication and assignation is not presented sufficiently to be credible.

Response: The compounds in Table 1 are exactly as described in the text. The figures quality was improved.

R3 comment to Response In lines 300-303 “The major compounds were: (-)-epicatechin (1), cyanidin 3-glucoside (2), procyanidin B2 (3), kaempferol-3-O-rutinoside (4), nobiletin (5), dihydrokaempferol (6), dios-301 metin (7), 3-O-Methylquercetin or isorhamnetin (8) isoorientin (9), genistein-8-C-glucoside (10) and 302 apigenin 6,8-digalactoside (11).” In Table 1, for example the compound 1 is a derivate of luteolin, derivative of (-)-epicatechin is a compound 3 in Table 1 and there is no (-)-epicatechin in Table1 and so on. The GNPS usage for dereplication and assignation is not presented sufficiently to be credible.

Response: We double checked the MS/MS spectra and GNPS library for these compounds. Dereplication and assignation was according 10.1016/j.funbio.2019.03.002

R3 At first, English should be corrected. For example (one of several), in lines 158- 159 „were treated with 0,25, 2,5, 25 and 158 250μg/ml OF WHAT? for 24, 48 and 72h. GM cultured cells were treated with 7,8 to 1000 μg/ml OF WHAT? for 24 h.“ Abbreviations should be defined eg GM , particularly in Abstract if they are not standard ones (e.g. GNPS). In Abstract, there is repetition and statements which have not been shown in the study (H2O2).

Response: We corrected this information.

R3  Figures 2 – 4 are very confusing and in figure captions it is not described enough what they present and how they should be interpreted.

Response: Corrected

R3 comment to Response Not enough to enable interpretation of the data obtained by GNPS. The authors do not refer within Results and Discussion to the figures 2-4.

Response: Discussion provides information about GNPS results (Lines 876-932)

In Figure 10, it is not clear what is on y-axis based on the Figure caption.

Response: Corrected

R3 comment to Response

In Figure 10 , by, for example, black colour they are marked percentage of cells which went into apoptosis after each of 4 different treatment or?  It is also not clear sufficiently what were NC and PC and in which concentration.

In each of Figure captions it should be described what were PC and NC if there were used in the experiment.

Response: Positive control was DMSO (100ul) and negative control was just cells with DMEM as described in legend. As described in the legend, black color represents apoptosis, blue color necrosis and green color viable cells.